# Global Globin Network Consensus Paper: Classification and Stratified Roadmaps for Improved Thalassaemia Care and Prevention in 32 Countries

**DOI:** 10.3390/jpm12040552

**Published:** 2022-03-31

**Authors:** Bin Hashim Halim-Fikri, Carsten W. Lederer, Atif Amin Baig, Siti Nor Assyuhada Mat-Ghani, Sharifah-Nany Rahayu-Karmilla Syed-Hassan, Wardah Yusof, Diana Abdul Rashid, Nurul Fatihah Azman, Suthat Fucharoen, Ramdan Panigoro, Catherine Lynn T. Silao, Vip Viprakasit, Norunaluwar Jalil, Norafiza Mohd Yasin, Rosnah Bahar, Veena Selvaratnam, Norsarwany Mohamad, Nik Norliza Nik Hassan, Ezalia Esa, Amanda Krause, Helen Robinson, Julia Hasler, Coralea Stephanou, Raja-Zahratul-Azma Raja-Sabudin, Jacques Elion, Ghada El-Kamah, Domenico Coviello, Narazah Yusoff, Zarina Abdul Latiff, Chris Arnold, John Burn, Petros Kountouris, Marina Kleanthous, Raj Ramesar, Bin Alwi Zilfalil

**Affiliations:** 1Malaysian Node of the Human Variome Project Secretariat, School of Medical Sciences, Universiti Sains Malaysia, Health Campus, Kubang Kerian 16150, Kelantan, Malaysia; halimfikri@gmail.com (B.H.H.-F.); shantysharadzin@yahoo.com (S.-N.R.-K.S.-H.); wardahyusofyaacob@gmail.com (W.Y.); 2Molecular Genetics Thalassaemia Department, The Cyprus Institute of Neurology & Genetics, 6 Iroon Avenue, Ayios Dometios, Nicosia 2371, Cyprus; lederer@cing.ac.cy (C.W.L.); coraleas@cing.ac.cy (C.S.); petrosk@cing.ac.cy (P.K.); marinakl@cing.ac.cy (M.K.); 3Faculty of Medicine, Universiti Sultan Zainal Abidin, Kuala Terengganu 20400, Terengganu, Malaysia; atifameen01@gmail.com; 4School of Health Sciences, Universiti Sains Malaysia, Health Campus, Kubang Kerian 16150, Kelantan, Malaysia; assyuhada@student.usm.my (S.N.A.M.-G.); nnorliza69@gmail.com (N.N.N.H.); 5Department of Paediatrics, School of Medical Sciences, Universiti Sains Malaysia, Health Campus, Kubang Kerian 16150, Kelantan, Malaysia; dianarashid4@gmail.com (D.A.R.); tihah1112@gmail.com (N.F.A.); sarwany@usm.my (N.M.); 6Thalassemia Research Centre, Institute of Molecular Biosciences, Mahidol University, Nakhom Pathom 73170, Thailand; grsfc@mahidol.ac.th; 7Department of Biomedical Sciences, Medical Genetics Research Centre, Faculty of Medicine, Universitas Padjadjaran, Bandung 40161, Indonesia; ramdan2749@gmail.com; 8Institute of Human Genetics, National Institutes of Health, University of the Philippines, Manila 1000, Philippines; ctsilao@up.edu.ph; 9Department of Pediatrics, College of Medicine, University of the Philippines, Manila 1000, Philippines; 10Department of Paediatrics & Thalassaemia Centre, Siriraj Hospital, Mahidol University, Bangkok 10700, Thailand; vip.vip@mahidol.ac.th; 11UKM Specialist Children’s Hospital, Jalan Yaacob Latif, Bandar Tun Razak, Cheras 56000, Kuala Lumpur, Malaysia; norunaluwar@ppukm.ukm.edu.my; 12Haematology Unit, Cancer Research Centre, Institute for Medical Research, National Institutes of Health, No. 1, Jalan Setia Murni U13/52, Seksyen U13, Bandar Setia Alam, Shah Alam 40170, Selangor Darul Ehsan, Malaysia; norafizayasin@gmail.com (N.M.Y.); ezaliaazlan@gmail.com (E.E.); 13Department of Haematology, School of Medical Sciences, Universiti Sains Malaysia, Health Campus, Kubang Kerian 16150, Kelantan, Malaysia; rosnahkb@usm.my; 14Hospital Ampang, Jalan Mewah Utara, Taman Pandan Mewah, Ampang Jaya 68000, Selangor, Malaysia; veena_263@yahoo.com; 15Division of Human Genetics, National Health Laboratory Service (NHLS) and School of Pathology, Faculty of Health Sciences, The University of the Witwatersrand, Watkins Pitchford Building, NHLS Braamfontein, Cnr Hospital and De Korte St, Hillbrow, P.O. Box 1038, Johannesburg 2000, South Africa; amanda.krause@nhls.ac.za; 16Nossal Institute for Global Health, MDDHS, University of Melbourne, Melbourne, VIC 3010, Australia; hmro@unimelb.edu.au; 17Global Variome, Institute of Genetic Medicine, International Centre for Life, Central Parkway, Newcastle upon Tyne NE1 3BZ, UK; julia@variome.org; 18Department of Pathology, Faculty of Medicine, Universiti Kebangsaan Malaysia Medical Centre, Jalan Yaacob Latif, Bandar Tun Razak, Cheras 56000, Kuala Lumpur, Malaysia; zahratul@ppukm.ukm.edu.my; 19Medical School, Université Paris Diderot, 75018 Paris, France; jacques.elion@inserm.fr; 20Clinical Genetics Department, Human Genetics and Genome Research Institute, National Research Centre, Cairo 12622, Egypt; ghadaelkamah@hotmail.com; 21Laboratorio di Genetica Umana, IRCCS Istituto Giannina Gaslini, Largo Gerolamo Gaslini 5, 16147 Genova, Italy; domenicocoviello@gaslini.org; 22Advanced Medical and Dental Institute, Universiti Sains Malaysia, Bertam, Kepala Batas 13200, Pulau Pinang, Malaysia; narazah@usm.my; 23Department of Paediatrics, Faculty of Medicine, Universiti Kebangsaan Malaysia Medical Centre, Cheras 56000, Kuala Lumpur, Malaysia; zarinaal@ppukm.ukm.edu.my; 24BioGrid Australia, Hodgson Associates, 4 Hodgson St., Kew, Melbourne, VIC 3101, Australia; chris@hodgsonassoc.com.au; 25Translational and Clinical Research Institute, International Centre for Life Times Square, Newcastle upon Tyne NE1 3BZ, UK; john.burn@newcastle.ac.uk; 26Department of Pathology, University of Cape Town City of Cape Town, Cape Town 7925, South Africa; raj.ramesar@uct.ac.za; 27Human Genome Centre, School of Medical Sciences, Universiti Sains Malaysia, Health Campus, Kubang Kerian 16150, Kelantan, Malaysia

**Keywords:** Global Globin Network, haemoglobinopathy, thalassaemia, low- and middle-income countries, epidemiology, disease burden, prevention program, Hemoglobinopathy VCEP, Human Variome Project

## Abstract

The Global Globin Network (GGN) is a project-wide initiative of the Human Variome/Global Variome Project (HVP) focusing on haemoglobinopathies to build the capacity for genomic diagnosis, clinical services, and research in low- and middle-income countries. At present, there is no framework to evaluate the improvement of care, treatment, and prevention of thalassaemia and other haemoglobinopathies globally, despite thalassaemia being one of the most common monogenic diseases worldwide. Here, we propose a universally applicable system for evaluating and grouping countries based on qualitative indicators according to the quality of care, treatment, and prevention of haemoglobinopathies. We also apply this system to GGN countries as proof of principle. To this end, qualitative indicators were extracted from the IthaMaps database of the ITHANET portal, which allowed four groups of countries (A, B, C, and D) to be defined based on major qualitative indicators, supported by minor qualitative indicators for countries with limited resource settings and by the overall haemoglobinopathy carrier frequency for the target countries of immigration. The proposed rubrics and accumulative scores will help analyse the performance and improvement of care, treatment, and prevention of haemoglobinopathies in the GGN and beyond. Our proposed criteria complement future data collection from GGN countries to help monitor the quality of services for haemoglobinopathies, provide ongoing estimates for services and epidemiology in GGN countries, and note the contribution of the GGN to a local and global reduction of disease burden.

## 1. Introduction

Haemoglobinopathies are recognised as a health burden in 71% of 229 countries, which together account for 89% of global births [1]. The prevalence of haemoglobinopathy carriers is around 5.2% of the world population, with over 330,000 affected infants born annually, comprising 83% of children with sickle cell disorders (SCD) and 17% of children with thalassaemia. In addition, haemoglobinopathies account for 3.4% of deaths in children under five years old [1].

Thalassaemia is a monogenic disorder with diverse clinical severities, partly owing to allelic heterogeneity at the alpha-globin and beta-globin loci and partly owing to modifier genes [2]. Moreover, thalassaemia syndromes occur with a high frequency in ethnic groups the origins of which can be traced to areas originally affected by malaria, such as countries in the Middle East, in Southeast Asia, and bordering the Mediterranean Sea [3]. Globally, 5.2% of the population is affected by some globin gene variation, whereas some regions or ethnic groups substantially exceed that frequency. These genetic and epidemiological complexities are addressed by databases on the ITHANET Portal covering globin and modifier variants and distribution, which helps to expand our knowledge of the genetic and molecular basis for phenotypic variation in haemoglobinopathies [4]. Meanwhile, inclusion of additional data on pertinent healthcare policies and organisations acknowledges the urgent need to chart, standardise, and improve local and global healthcare systems, to achieve a higher standard of haemoglobinopathy diagnosis, prognosis, counselling, care, and therapy.

As thalassaemia syndromes emerged as a global health burden [5,6], they came into the focus of the international non-governmental organisation (NGO), the Human Variome Project (HVP) [7]. The HVP strives towards achieving a significant reduction in the genetic disease burden on the world’s populations [8]. HVP aims to ensure that all information on genetic variation can be collected, curated, interpreted, and shared freely and openly, for faster, better, and cheaper diagnosis and management of genetic diseases [9]. Additionally, better insight into their causes and severity is envisioned [10]. HVP is organised in country nodes, which facilitates the sharing of information on human genetic variation [11], and supports the vision of a world where the availability of, and access to, genetic variation information is no longer a limiting factor for diagnosis, prognosis, and treatment. It is hoped that interdisciplinary and intercultural collaboration will produce better and cheaper therapies for genetic diseases. Following its successful initiation to the BRCA Challenge, a network focusing on breast cancer and on strengthening data exchange for existing databases mainly in developed countries [12], the HVP turned its attention to haemoglobinopathies as a corresponding challenge in low- and middle-income countries (LMICs) in 2015, initiating the Global Globin 2020 Challenge [13], which was renamed the Global Globin Network (GGN) in 2020. A key GGN milestone in collaboration with ITHANET was establishing the ClinGen Hemoglobinopathy Variant Curation Expert Panel (VCEP) for the curation of knowledge on haemoglobin variants [14,15], and their classification into American College of Medical Genetics/Association for Molecular Pathology (ACMG/AMP) categories from benign to pathogenic [16,17], the output of which will be instrumental to realising accelerated and more reliable diagnosis and prognosis of haemoglobin disorders based on high-throughput technology [15,18].

The GGN was established to address key development areas related to haemoglobinopathies in LMICs, with the notion that a focus on LMIC-relevant diseases such as SCD and thalassaemia would, first, incentivise the adoption of current technological advances in human genomics to systematically collect and share genetic variation data, and would, second, pave the way towards a wider application of corresponding technologies and procedures to other diseases [19]. The GGN seeks to improve the quality and quantity of curated genomic data contributed by LMICs to internationally recognised genetic databases, in line with international best practices and in adherence with applicable ethical and regulatory frameworks and policies that assist and protect patients. At the same time, biotechnical systems and procedures are being developed, and exchange of expertise within the GGN will support LMICs with much-needed infrastructure and expertise development, as a starting point leading onto similar changes in other areas of public health service provision and for synchronised data-sharing between countries [20].

Organised by HVP country nodes and headed by Professor Zilfalil Bin Alwi (Malaysia) and Professor Raj Ramesar (South Africa), the GGN is supported by a steering committee that consists of representatives from 32 country nodes. The network comprises professional stakeholders in haemoglobinopathies, including clinicians, researchers, curators, bioinformaticians, counsellors, patient groups, and health administrators. Given the interdisciplinarity of the GGN consortium members and the wide divergence of the GGN’s participating countries in their geographical area, economic status, culture, and religious practice, the GGN is uniquely placed to analyse countries by their approaches to thalassaemia management and prevention.

## 2. Materials and Methods

Based on the expert opinions and discussion during annual GGN meetings [13], four country categories have been devised to support the creation of meaningful summary statements and roadmaps for further development, irrespective of other divergent societal factors. Based on data collected by the GGN, accessible on the ITHANET Portal in interactive databases for haemoglobin variations and epidemiology [4], this article aims to describe the features and characteristics of the countries grouped into four categories, and will make corresponding recommendations for experts on database development and disease management in alignment with the existing databases. The four categories are based on ITHANET data and variables, and a division of variables into two main groups: qualitative and quantitative. We have further characterised nine “major” and two “minor” variables within qualitative data according to expert opinions and the incidence/prevalence of specific variables within different countries (Table 1). 

Major qualitative variables were then considered as the scoring criteria for individual grouping of countries within four categories, A to D. Importantly, all variables can be re-assessed over time to re-categorise GGN member countries based on changing parameters. The overall methodology used in this study from data collection to GGN classification is shown in Figure 1.

The resulting GGN classification is based on accumulative scoring indicated by two identifiers, i.e., a letter for categories A to D and a superscript number representing the number of minor qualitative variables applicable (Table 2). Accordingly, the accumulative score A^1^ for Cyprus would indicate a category-A country with MRI facilities to guide disease management. Similarly, Italy is under category A with an accumulative score of A^1,2^, reflecting that it is a category-A country with all the major and minor qualitative variables. By contrast, Myanmar, with an accumulative scoring of D^0^ is under the category D, i.e., without listing minor qualitative variables on ITHANET. Quantitative variables were not used for categorisation, as we did not intend to base future predictions and estimates on inferential statistics.

## 3. Results

This section may be divided by subheadings and provides a concise, precise description of the experimental results, their interpretation, and the experimental conclusions that can be drawn.

All GGN member countries were analysed, and data were curated from ITHANET (Appendix A). We evaluated all the variables as per our criteria and listed all GGN member countries accordingly, also giving the accumulative scores for each country (Table 2). It was found that only six countries, specifically Cyprus, France, Italy, Malaysia, Singapore, and the United Kingdom, were listed as per our criteria in category A, with differences in minor qualitative variables, which are specified throughout as superscripts. Sixteen countries were listed after analysis in group B: Australia, Bangladesh, Belgium, China, India, Indonesia, Iran, Netherland, Nigeria, Pakistan, Portugal, South Africa, Spain, Thailand, Turkey, and Vietnam. Similarly, seven countries—Brazil, Cambodia, Egypt, Nepal, Philippines, Democratic Republic of the Congo, and Sri Lanka—were listed in category C. Finally, category D comprised a total of three countries, specifically Brunei, Laos, and Myanmar. All the categories are elaborated further in this study, based on reference to variables laid out in Table 1 (Appendix A), as available on the ITHANET Portal and in peer-reviewed publications (Table 3, Table 4, Table 5 and Table 6).

### 3.1. Category A

Countries in GGN category A have comprehensive prevention programmes. They have accomplished zero or low birth rates of thalassaemia despite a high carrier prevalence. Category A is represented in the consortium by Cyprus, France, Italy, Malaysia, Singapore, and the United Kingdom (Table 3). Singapore and Cyprus have birth rates below 1%, despite national carrier frequencies of α- and β-thalassaemia, respectively, of 20% and 12% for Cyprus and 4% and 3% for Singapore [1,21,22,23]. This has freed up funds for the optimised management of existing patients, resulting, e.g., in average ages of the patient population in the 40–50s and birth rates for thalassaemic women rivalling those of the general population [24,25,26]. Conversely, the estimated thalassaemia birth rate in these countries has decreased gradually every year, up to a point where improved disease management has resulted in a steady-state rate of voluntary thalassaemia births. This latter point represents a level of empowerment of patients and carriers in GGN A countries that cannot currently be attained in countries in categories B to D, where the highest level of possible disease prevention is a key aim of all disease control efforts. GGN A countries have also largely transformed patient care for thalassaemia from paediatric to adult, with a corresponding shift in disease-associated complications.

Since the late 1970s, adult or premarital population-screening programmes, genetic counselling, and prenatal diagnosis have been introduced among the at-risk populations in the Mediterranean populations, including Sardinians, continental Italians, Greeks, and Cypriots. All prevention programmes were highly successful in disease prevention, resulting in significant cost savings without compromising patient care, and additionally achieving thalassaemia-related education of at-risk populations and health professionals. According to Cao et al. (2002), there has been a significant decrease in the thalassaemia major birth rate in all the Mediterranean countries where education and counselling were introduced. For example, in the Sardinian population in Italy, the birth rate of thalassaemia major has declined from 1:250 to 1:4000 live births [27].

In Cyprus, the Cyprus Thalassaemia Clinics and Screening Laboratory have been conducting population screening for haemoglobinopathies since 1978. A vital pillar of the prevention programme is still a relatively high ethnic homogeneity of Cypriot society combined with pre-marital screening, where issuance of marriage certificates by the Greek-Orthodox church is contingent upon testing (but not the test outcome) for the thalassaemia disease or the carrier status of both partners. Samples from individuals with abnormal haematological indices are sent centrally to the Cyprus Institute of Neurology and Genetics (CING), to the Molecular Genetics Thalassaemia Department, for molecular characterisation and identification of haemoglobinopathy carriers. Additional individuals have been referred for molecular analysis as part of family studies, and genetic analysis has been performed as part of a prenatal diagnosis for couples at risk of births affected by thalassaemia [23]. As a result, the Cypriot prevention programme has achieved the most globally significant difference between expected and actual thalassaemic births. Based on its pioneering role in thalassaemia disease prevention, CING hosts and develops the ITHANET community portal, coordinates the International Hemoglobinopathy Research Network (INHERENT) [28], and is the central database developer for the Rare Anaemia Disorders European Epidemiological Platform (RADeep; https://www.radeepnetwork.eu/, last accessed 3 January 2022), an initiative of the European Reference Network for rare haematological diseases (ERN-EuroBloodNet). The success of the thalassaemia programme and the active role of patient organisations in Cyprus have contributed to broader public awareness of thalassaemia syndromes and of the required blood donations, pre-marital screening, and genetic counselling for couples at risk. Additionally, dedicated clinicians in each city are involved in thalassaemia disease management, and annual national thalassaemia conferences and patient society conferences keep all stakeholders informed of the newest developments in therapy, diagnosis, and disease management.

In Singapore, the National Thalassaemia Registry (NTR) was established in 1992, located at the Kandang Kerbau (KK) Women and Children’s Hospital [29]. The aim of the NTR is to register and give free counselling to all thalassaemia carriers. According to Ng et al. (1994), screening family members of index cases has effectively identified new carriers. Furthermore, the rate of thalassaemia major has decreased significantly with adequate, effective screening and preventative strategies [30]. Early prenatal diagnosis of thalassaemia in the first trimester allows early termination of a thalassaemic foetus as one of the preventative measures in this country [31]. A study by Kow Yin et al. (2004) found that the most effective method of screening among the local population is molecular screening, which determines 9% of thalassaemia carriers, compared to phenotype screening, which only identifies 6% [32].

Meanwhile, in Malaysia, the Malaysian Ministry of Health initiated a national prevention and control programme in 2004. The prevention programme was intensified by screening all school students in 2014 to identify the prevalence of thalassaemia, and thalassaemia awareness campaigns were conducted by government and non-governmental organisations. The Ministry of Health Malaysia (MOH) also established the Malaysian Thalassaemia Registry (MTR) in 2007, which continues to aggregate data from all participating hospitals under the MOH and from university hospitals under the Ministry of Higher Education (MOHE). The MTR is the first online patient registry in Malaysia equipped with real-time data entry, updates, and data reports. It allows enrolled users to observe the aggregated data at any point in time. This registry holds detailed epidemiological and clinical data of patients that provide an understanding of thalassaemia trends in Malaysia. Based on this database, the percentage of thalassaemia carriers in Malaysia is understood to be 6.8% of Malaysians, and the total number of thalassaemia patients registered in Malaysia was 8681 at the time of assessment [33,34]. The effort to collect thalassaemia data in this country was also supported by the Malaysian Node of the HVP (MyHVP), which established a locus-specific database (LSDB) to collate and curate genotype and phenotype thalassaemia data from published scientific papers. The HVP Southeast Asia Node will expand this effort in the future by establishing an LSDB on thalassaemia as part of efforts to combat the most common genetic disease in this region using data-driven approaches.

The United Kingdom has long had a large immigrant community for both SCD- and β-thalassaemia-endemic populations and has, accordingly, put dedicated prevention measures in place. These include premarital and preconception screening upon request, with coverage of prenatal screening by the National Health Service, along with antenatal screening and SCD screening, across all parts of the United Kingdom. After establishing the European Haemoglobinopathy Registry (EHR) at Central Middlesex Hospital in 2004, a dedicated National Haemoglobinopathy Registry (NHR) was established in 2009 for both β-thalassaemia and sickle cell disease, in support of the national disease prevention and management effort for haemoglobinopathies [21,35,36,37].

**Table 3 jpm-12-00552-t003:** Summary of thalassaemia prevention policies in GGN countries in category A.

Country	Prevalence of Carriers	Ethnic Group(s) Affected	Policy Applied in the Country				
			Prevention Programme		Prenatal/Antenatal Screening	Sickle Cell Disease Newborn Screening	Thalassaemia Registry
			National Level	Local/District Level			
Cyprus	**βThal** 12% (Greek Cypriots) **αThal** 20% (Greek Cypriots) [23,38,39]	Greek Cypriot Turkish Cypriot [23,39]	Yes [23,39]	Yes [23,39]	Yes [23,39]	No	National Thalassaemia Registry [39]
France	**βThal** (0.7%) **SCD** (0.7%) **HbE** (0.15%) **HbC** (0.2%) [40,41]	Guadeloupe			No [21]		Yes [21]
Italy	**βThal** (6%) **αThal** (6%) **SCD** (2%) **HbC** (1%) **HbE** (0.2%)	Italian, Sardinian [27]	Yes [21]	Yes [21]	Yes [21,27]	Yes [21]	National Thalassaemia Registry [21,27]
Malaysia	**βThal** (4.5%), **αThal** (4.9%), **HbE** (5.5%) [42,43]	Malay (62.0%) Kadazan-Dusun (14.0%) Chinese (13.0%) Indian (1.0%) Others (10.0%) [33,34]	Yes [33]	Yes	Yes [42]	No	Yes [Malaysian Thalassaemia Registry] [33,34]
Singapore	**βThal** (1.6%), **αThal** (incl. HbCS) (5.5%) **HbE** (1.7%) [44]	αThal/βThal Chinese (6.4%/2.7%) Malay (4.8%/6.3%) Indian (5.2%/0.7%) [44]			Yes [44]		National Thalassaemia Registry [29]
United Kingdom	**βThal** (0.44%) **SCD** (2.5%) **αThal** (2.5%) **HbC** (0.13%) [40]	Irish, Anglo-Saxon and multi-ethnic	Yes [35]		Yes [21,36]	Yes [35]	Yes [37]

Abbreviations: αThal—alpha-thalassaemia, βThal—beta-thalassaemia, SCD—sickle cell disease, HbCS—Hb Constant Spring. Major carrier categories are shown in bold to aid understanding. Unless otherwise indicated, data are from IthaMaps [4,45].

### 3.2. Category B

GGN category B countries show efforts to establish a national control programme but have limited availability/accessibility of related services for patients and carriers. Category B is represented in the consortium by Australia, Bangladesh, Belgium, China, India, Indonesia, Iran, Netherland, Nigeria, Pakistan, Portugal, South Africa, Spain, Thailand, Turkey, and Vietnam (Table 4). While most of these countries report a national control programme, not all elements of such programme are implemented, including raising awareness among families and patients, health professionals, and the community in general, as well as improving access to services, establishing national centres of expertise to provide advice, and ensuring that savings from disease prevention are returned to expand and improve services [13].

Thailand started a national programme in 1997 to prevent and control homozygous β-thalassaemia, compound heterozygous β-thalassaemia/HbE, and Hb Bart’s hydrops fetalis. The programme includes participation by public health offices, including the government departments of medical services, health, and medical sciences. This programme is cost-free. Pregnant women were the first target group for thalassaemia screening based on the osmotic fragility test. The husband was called in for screening, too, if his wife’s result was positive. If both were positive, a standard method of applying a confirmation test for thalassaemia was implemented. Yet, 20 years after launching the nationwide control programme for severe thalassaemia, still, only an approximately 50% coverage of preventative screening for pregnant women had been achieved because many women attended the antenatal care clinic late [22,46].

In China, a prevention programme was launched in Guangxi in 2010, prompted by the high carrier rate of thalassaemia and HbE in this area. Thalassaemia screening is conducted at marriage registration centres, where newly married couples are educated about birth control, human immunodeficiency virus, and other issues, including thalassaemia, and where blood samples are taken and sent to the hospital for further analyses. Couples are then informed of positive results in preconception counselling, followed by prenatal diagnosis of potentially affected pregnancies. Within five years of running this programme, approximately 50% of pregnant women were then being screened for thalassaemia, leading to a substantial reduction of births affected by thalassaemia major and intermedia [22].

In India, the incidence of births with thalassaemia is 10,000 cases every year, equating to 10% of the total world incidence of thalassaemia-affected children. The prevalence of thalassaemia ranges between 0.6% and 15% across south India [47].

**Table 4 jpm-12-00552-t004:** Summary of thalassaemia prevention policies in GGN countries in category B.

Country	Prevalence of Carriers	Ethnic Group(s) Most Affected	Policy Applied in the Country				
			Prevention Programme		Prenatal/Antenatal Screening	Sickle Cell Disease Screening	Thalassaemia Registry
			National Level	Local/District Level			
Australia	**βThal** (0.4%) **HbE** (0.4%) [22,48]			Yes [22]	Prenatal (Yes), Antenatal (No) [22]		Yes
Bangladesh	**α/βThal** (4.1–12%) **HbE** (6.1%) [49,50]	Bengali, Marma, Khyang	No		No	No	
Belgium	**SCD** (0.42%) **βThal** (0.28%) **HbE** (0.02%) **HbC** (0.02%) [40,41,51]	Northern European (lowest risk)	No [35]		No	No [51]	Yes
China, Guanxi	**αThal** (3.54%) **βThal** (6.78%) **HbE** (0.42%) [22]			Yes (Regional) [35]	Yes [52]	No [35]	Yes [53]
India	**αThal** (41.0%) **βThal** (3.9%) **HbE** (1.0%) [22] 1:8 of all Thal carriers worldwide; regional range 0.6–15% [47]	Gujarat (10–15%), Tamil Nadu (8.5%), Punjab (6.5%) [47]		Yes [52]	Yes [52,54]	Yes [35]	Yes [52]
Indonesia	**αThal** (10.9%) **βThal** (5%) **HbE** (6%) [22]	Malay, Javanese, Aceh, Batak, Sundanese, Padang, Betawi, South Celebes		Yes [22]	Yes [22]		Yes (National)
Iran	**βThal** (6%) **SCD** (1%) **αThal** (30%) [40,55]		Yes (National) [56]		Yes [56]	No [35]	Yes [57]
Netherlands	**SCD** (0.18%) **βThal** (0.4%) **HbE** (0.07%) **HbC** (0.1%) **αThal** (3.6%) [40,41]	Dutch		Yes [35]	Prenatal (No) [58] Antenatal (Yes) [21]	Yes	Yes [21]
Nigeria	**SCD** (25.0%) [59]	Yoruba	Yes		Yes	Yes	No
Pakistan	**βThal** (5%) **αThal** (2.4%) **SCD** (0.27%)			Yes	Yes		
Portugal	**βThal** (1.63%) **SCD** (0.12%) **HbE** (0.002%) **HbC** (0.01%) [41,60]	Portuguese	No [35]		No	No [35]	
South Africa	**βThal** (2–20%) Indian/Mediterranean **SCD** (≤20%) Central and West African **αThal** (≤30%) single alpha deletion	Mediterranean, Indian, Central and West African	No	No	Yes	No	No
Spain	**βThal** (1.64%) **SCD** (0.3%) **HbE** (0.002%) **HbC** (0.03%) [40,41]	Spanish	No [35]		No [35]		Yes [37]
Thailand	**βThal** (3–9%) **HbE** (10–53%) **HbCS** (1–8%) **αThal** (20–30%) **HbE** (33% [22]	Thais [22]	Yes	-	Yes [22,52]	No	Yes
Turkey	**βThal** (2.2%) **SCD** (0.44%) **αThal** (2%) **HbE** (0.002%) **HbC** (0.001%) [40]	Turkish	Yes [61]	-	Yes [61]	No [35]	Yes [62]
Vietnam	**αThal** (11.7%) **βThal** (2.6%) **HbE** (1%) [22]	Kinh Muong Tay		Yes			No [63]

Abbreviations: αThal—alpha-thalassaemia, βThal—beta-thalassaemia, SCD—sickle cell disease, Hb-CS—Hb Constant Spring. Major carrier categories are shown in bold to aid understanding. Unless otherwise indicated, data are from IthaMaps [4,45].

### 3.3. Category C

The GGN countries in this category are Brazil, Cambodia, Egypt, Nepal, the Philippines, the Democratic Republic of the Congo, and Sri Lanka (Table 5). These countries have substantial expertise in the diagnosis, treatment, management, and prevention of haemoglobinopathies. However, these measures are not part of a sustainable national control programme, and the national thalassaemia birth rate is still close to that predicted by carrier rates. Nevertheless, some countries have made progress in improving the health status of their patients against a backdrop of high birth rates for thalassaemia and have set up local or regional control programmes that might serve as hubs for wider regional or national expansion of disease prevention.

In Pakistan, for example, the level of consanguinity is very high in the community, and only cascade screening, i.e., the targeted genetic testing of families of index cases, is performed to identify the high prevalence of carriers [64,65]. This strategy effectively preserves resources while raising awareness of the disease and allowing genetic counselling of couples at risk, which is expected to reduce the frequency of β-thalassaemia in Pakistan [66]. Based on the ITHANET portal, this country does not maintain a patient registry or provide dedicated treatment centres for thalassaemia [67]. Although Pakistan has launched a thalassaemia prevention programme, the number of thalassaemia patients is still high, owing to a lack of facilities for thalassaemia patient management and lack of awareness. Taken together, these factors have limited progress towards reducing the disease burden of thalassaemia in this country.

In Sri Lanka, the National Prevention Thalassaemia Program was launched in 2006, through which the country has reached a commendable standard of care and disease prevention. However, beyond good basic care and successful screening and counselling in cases of thalassaemia, Sri Lanka still needs to improve its clinical and laboratory services for a patient population whose prolonged survival requires increasingly complex care [68]. This country also needs a national database to ensure the effectiveness of its screening programme, as the documentation of β-thalassaemia major is still incomplete and, to date, no apparent reduction in the number of thalassaemia patients has been achieved since the launch of the national prevention programme [69].

**Table 5 jpm-12-00552-t005:** Summary of thalassaemia prevention policies in GGN countries in category C.

Country	Prevalence of Carriers	Ethnic Group(s) Most Affected	Policy Applied in the Country				
			Prevention Programme		Prenatal/Antenatal Screening	Sickle Cell Disease Screening	Thalassaemia Registry
			National Level	Local/District Level			
Brazil	**βThal** (6%) **SCD** (1%) **αThal** (30%) [40,70]	Brazilian	No			Yes [71]	Yes
Cambodia	**βThal** (0.18%) **αThal** (18.27%) **HbE** (19.93%) [72]	Khmer with regional differences	No		No	No	
Egypt	**βThal** (5.3%) **SCD** (2.54%) **αThal** (9.25%) [40,73]		Yes	No	Yes [52]	No [35]	
Nepal	**HbE** (4.0%) [74]	High case counts in Bheri Zonal Hospital, Nepalgunj; low in Bharatpur Hospital, Chitwan	No				
Philippines	**αThal** (20.4%) **βThal** (1.2%) **HbE** (0.4%) [22]		No	No	No	Yes	No
Democratic Republic of the Congo	**SCD** (24%, inferred from 2% homozygote births) **αThal** (1% non-deletional, inferred from 49% carriers among SCD patients) [75,76]		No				
Sri Lanka	**αThal** (6.5%) **βThal** (2.5%) **HbE** (2.5%) [22]		Yes [52]	Yes	Yes [77]		

Abbreviations: αThal—alpha-thalassaemia, βThal—beta-thalassaemia, SCD—sickle cell disease. Major carrier categories are shown in bold to aid understanding. Unless otherwise indicated, data are from IthaMaps [4,45].

### 3.4. Category D

Countries such as Brunei, Laos, and Myanmar (Table 6) fall within this category. Characteristics include an overall lack of information about thalassaemia carrier rates, affected births, and prevention efforts. Affected countries fall into the low-income country category, where health services are minimal. The challenges of thalassaemia control in these countries include an overall lack of resources for medical services, substantial expenditure for other health challenges, including infectious diseases, and specifically for thalassaemia, a lack of (i) policies for disease control, (ii) national guidelines for thalassaemia screening, (iii) standardised treatment protocols, and (iv) human resources for dedicated basic thalassaemia care and prevention, such as haematologists and laboratory technicians. Additionally, infrastructural problems include a lack of suitable medical equipment and exclusion of iron chelators from the essential drug list, leaving them unavailable to thalassaemia treatment units. In the majority of countries in this category, other critical elements of prevention programmes are also absent or only available at a highly localised level, including genetic counselling, structured population screening, premarital counselling, and prenatal diagnosis [78].

Lack of policy, personnel, and infrastructure creates a lack of thalassaemia disease information, such as data concerning national, ethnic, or regional mutation spectra and their heterogeneity in the population. Information on mutation frequencies and distribution is essential for any data-guided policies and for establishing regional or national control programmes, such as those fully implemented in GGN category-A countries.

Severe health challenges and costs of thalassaemia and other unchecked diseases, the absence of data to guide prioritised health spending, and typically low national health budgets are among the further contributing factors. In the absence of any prevention programme, high thalassaemia birth rates with unmet blood transfusion and chelation requirements are combined with a continuous vicious cycle of spiralling case numbers, management costs, and disease-related mortality. This cycle might be impossible to break without external advice or intervention. To assist these countries in managing thalassaemia and addressing the absence of a nationwide programme, initiatives from NGOs, such as the Thalassaemia International Federation (TIF), and patient-based societies, and bi- or multilateral collaboration between Asian countries have proven helpful. For instance, experts from Thailand have been working with their colleagues in Bangladesh, Cambodia, Myanmar, Laos, and Vietnam to identify molecular abnormalities of both α-thalassaemia and β-thalassaemia endemic to the Asian region, and training programmes to help train technicians and scientists from these countries have existed for quite some time [79]. Such efforts greatly benefit from support and coordination by international agencies. They are critically required to lift GGN category-D countries into category C, reduce the national health burden, and lessen the suffering of the mostly paediatric patient population.

**Table 6 jpm-12-00552-t006:** Summary of thalassaemia prevention policies in GGN countries in category D.

Country	Prevalence of Carriers	Ethnic Group(S) Most Affected	Policy Applied in the Country				
			Prevention Programme		Prenatal/Antenatal Screening	Sickle Cell Disease Screening	Thalassaemia Registry
			National Level	Local/District Level			
Brunei	**αThal** (4.3%) **βThal** (2%) **HbE** (0%) [22]	Malay	No		No	No	Yes
Laos	**αThal** (26.8%) [80] For pregnant Laotians: **HbE** (30.1%) **α^0^Thal** (8.6%)		Yes		Yes	Yes	
Myanmar	**αThal** (10–56.9%) **βThal** (0.5–4.1%) [78]		No				

Abbreviations: αThal—alpha-thalassaemia, βThal—beta-thalassaemia, SCD—sickle cell disease. Major carrier categories are shown in bold to aid understanding.

## 4. Discussion

In this paper, we have laid out readily assessable qualitative parameters, supplemented with carrier frequencies as quantitative parameters, for the 32 member countries of the GGN, which allows their separation into four actionable categories of achievement in the management of haemoglobinopathies (see Figure 2). Our data curation of GGN member countries through ITHANET revealed factors that, in their totality, give a clear indication of each country’s efforts, progress, and success in disease management and prevention, and therefore, allow us to make stratified recommendations for the improvement of health provision.

### 4.1. Screening and Disease-Prevention Strategies

By definition, disease control is a key common feature for same-category countries and a distinguishing feature between categories. Disease control of thalassaemia requires regional or national policies for the identification of disease carriers and genetic counselling regarding their own risks and those of their potential offspring [1]; it is hoped that the introduction of such policies and their consistent implementation will lead to a decrease in births of affected children. Historically, such introduction occurred in three stages [81], in line with progress in developing the underlying technologies and achieving political and societal insights. The first stage is informing parents with affected children of their risk of thalassaemia at each subsequent birth, which may prompt them to limit the size of typically large families and thus reduce the thalassaemia prevalence at a regional or national level [65]. Second, the introduction of early prenatal diagnosis for couples with affected children may lead to a lower birth prevalence. Finally, providing prospective carrier screening for the whole population is beneficial in prompting early prenatal diagnosis, or the more advanced intracytoplasmic sperm injection combined with pre-implantation genetic diagnosis, even before the first affected child has been born to a family. The current state-of-the-art technology and international collaboration and training within the GGN consortium and beyond may help countries in lower categories leapfrog several of these steps and bypass intermediate stages of investment and implementation.

A key technology development for rural and economically disadvantaged areas would be affordable and instantaneous point-of-care testing. Currently, the Malaysian Node of the HVP, under Universiti Sains Malaysia, is collaborating with a UK-based biotechnology company, QuantuMDx Ltd. (Newcastle upon Tyne, UK), to develop a β-thalassaemia screening device based on a microfluidic microarray biosensor platform. The development of this point-of-care (POC) diagnostic device will simplify and lower the cost of haemoglobinopathy screening by directly applying genetic analysis to detect pathogenic variants that cause the disorders. The simplicity and rapidity of this POC testing process ensure it can be run with minimal training and in a single session for both sampling and communication of the test results. At the same time, its sensitivity and specificity enable clinicians to avoid using centralised and costly standard haematological and molecular diagnostic tests for a precise diagnosis. The development of this low-cost portable device aims to significantly further the affected countries’ socioeconomic development and lessen the burden on their healthcare system.

The strategy of carrier screening varies according to a few factors, such as pre-existing carrier data for couples at risk, social attitudes, coverage of cost by health authorities, and religious and legal frameworks in each country. High-school or pre-marital carrier screening is cost-effective and still allows for a full range of reproductive choices without the need for multiple tests over the individual’s lifetime, but relies on pre-existing carrier data for couples to avoid late detection of affected births. By contrast, newborn screening allows for early detection of affected individuals, where no carrier data for couples exist, but usually requires repeat testing of the adolescent or adult. Finally, screening during pregnancy gives fewer options, is ethical only if the prenatal diagnosis is freely available and the visit and timing depend on individual initiative, and often identifies the risk too late to provide individuals or couples with a full range of reproductive choices.

Population screening is not the only helpful strategy: family studies can be cost-effective where consanguineous marriage is common or in countries with a low carrier prevalence. The effects of thalassaemia screening depend on the choices made by informed individuals. Later, the birth prevalence of thalassaemia will decrease because most at-risk couples may limit their number of children, and some may opt to terminate the pregnancy; also, some carriers may avoid risk by selecting a non-carrier partner.

Early diagnosis and disease management are matters of policy but are also a function of disease screening and prevention strategies. A precise molecular diagnosis is often secondary to the initial screening results. Good disease management also presupposes that a sufficiently low number of patients allows for an adequate blood supply for transfusions and frees up resources to provide costly blood chelating agents. Early mortality owing to thalassaemia is a function of prevention, diagnostic, and management strategies in all categories.

Cost analysis reports can observe the economic burden of thalassaemia, but unfortunately, cost analysis data for GGN countries are still scarce. As shown in a report for Thailand [82], high carrier frequencies may bring a high economic burden to each country. Studies on the economic burden of β-thalassaemia have been reported from the United Kingdom [83], Canada, Israel [84], Taiwan, Sri Lanka [85], Myanmar [86], and Thailand. For Thailand, the direct medical cost was approximately 60% of the total cost [82]. In contrast, the total cost of taking care of affected children from a societal perspective, covering the direct medical costs, direct non-medical costs, and the indirect costs, was US $950 per patient annually [82]. As healthcare costs are generally increasing, ignoring its impact on healthcare providers becomes increasingly difficult. Attempting to contain costs, hospital systems are shifting the burden of managing diseases back onto patients and families, and publicly funded services are being rationed, disproportionately affecting poorer families.

Despite the implementation of prevention programmes in countries in GGN category B, there has been no decrease in the incidence of the disease. Such failure might be attributed to a lack of recognition of problems related to thalassaemia, suboptimal coordination of teamwork and services, lack of training and inadequate numbers of counsellors, ineffective thalassaemia support groups, and limited awareness of research in thalassaemia prevention and control, all of which can reduce the potential effectiveness of prevention programmes [87].

The success of screening in several countries with largely homogeneous endogenous thalassaemia carrier populations, such as Cyprus, Greece, and Sardinia/Italy, might have more to do with healthcare and counselling being implemented with and from within the community than with the efficiency of the health services concerned. Acceptance and trust in the advice and facts conveyed by clinicians and counsellors are critical determinants of the success of disease control programmes, which might be improved in the more homogeneous category-D countries to boost the success of any nascent prevention programmes.

In countries like Cyprus, Denmark, the Maldives, and Singapore, where the population is small and health services are well-funded and well-organised, screening a large fraction of the population is possible. By contrast, for larger populations and developing countries, such as India and Indonesia, screening the entire population would not be as practical. Hence, cascade screening might need to be implemented to allow for an acceptable cost/benefit ratio of a growing disease control programme. Innovative and technologically advanced implementation of screening by different strategies was the mainstay of the greatest thalassaemia prevention successes and the most effective decreases in affected birth rates. This was demonstrated in Cyprus with its compulsory premarital test certificates and availability of antenatal diagnosis from 1977, and in Singapore with its cascade screening strategy from 1980, along with antenatal diagnosis in 1990, which resulted in a high detection rate among high-risk sub-populations [88,89]. Smart screening relies on political will, finance, and infrastructure. Although government recommendations were made early on for non-compulsory screening programmes in many GGN countries, such as Malaysia, Myanmar, Thailand, and others, success was limited. Limited screening budgets do not allow for full penetration of high-risk populations in populous countries, thus limiting the national success of screening. In South Africa, national screening would not be appropriate as the conditions affect small groups within the population. It would be more effective if there was screening aimed at individuals with particular ancestries. This is not done at the national level, although individual families can access screening.

Finally, in countries and communities where cascade screening was applied, few patients came forward for analysis, owing to a lack of awareness of screening campaigns, which critically need to be backed up with education campaigns. Moreover, in many communities, the choice of abortion, even for early antenatal detection, was not an option for ethical or religious reasons (see below), leaving couples of carriers with the awareness of their risks but without the tools to address them. This effectively lowers the ability of identified carriers to find a spouse (“stigmatisation”), and thus, creates counter-incentives to participate in any screening programme. Data-based planning of screening, solid funding, and education, along with other factors that contribute to compliance and trust of the high-risk population, are thus critical for screening programmes.

### 4.2. Education and Public Awareness

Many countries have established local or national thalassaemia societies to raise public awareness. The presence of thalassaemia societies will be effective as a mediator in the community to create trust and disseminate information on thalassaemia. Social and mass media can be used as a medium to promote public awareness. This includes sharing-of-knowledge activities such as organising annual seminars and lectures. Parents’ associations can play an important role by providing financial assistance and support for thalassaemia management, providing psychological assistance to parents and patients, and galvanising the efforts of parents as an influential group [88].

A significant number of healthcare providers are not trained to discuss pregnancy termination options (abortion) and assume that thalassaemia, as a treatable disease, does not justify abortion. However, this is often based on misconceptions about the impact of thalassaemia and about the legal and religious grounds for abortion. Some abortions are allowed based on certain criteria and conditions [42]. In many GGN Category-B and -C countries, preventing antenatal death and decreasing the rate of thalassaemia births among detected carriers to ensure a decrease of the thalassaemia birth rate remains a challenge. One reason for this is the low number of participants who agree to undergo antenatal screening followed by termination of a positive thalassaemia pregnancy [42]. This results in many carrier parents effectively allowing recurrent births of thalassaemic children after a thalassaemic first-born child without taking preventive measures, often motivated by a desire to have a subsequent non-thalassaemic child and based on a lack of insight into the underlying inheritance pattern. This would be easier to quantify if recurrent thalassaemic births were well-recorded in the countries affected. Still, the trend is already apparent from the frequent observation of siblings from the same at-risk couple coming to thalassaemia centres for treatment.

Therefore, education and perception play into the decision process for at-risk pregnancies as much as legal concerns. A lack of knowledge and understanding often leads to general confusion over the severity of thalassaemia and the appropriateness of a termination. Finally, in many communities, religious considerations outweigh even legal concerns or education about thalassaemia and abortions. Reference to ‘fate’ by many women suggests fatalistic beliefs, which their Islamic faith may influence, and suggests that Muslim South Asians often use religion as a coping strategy. This might also be related to their social class, educational background, and religious belief. For example, in a sample of young people with β-thalassaemia major, it was a commonly held belief that ‘Allah would only give the illness to those who had the resources and strength to cope with it…’ [90]. Further research is needed on the role of fatalistic beliefs and how they impact the decision-making process for prenatal screening [91]. This can be addressed systematically and improve the proportion of people making an informed choice in counselling.

Recently, Ngim et al. (2013) reported that in Malaysia, with Islam as a major religion, religious belief was significantly associated with the decision to terminate an affected pregnancy among parents who agreed to an antenatal diagnosis [42]. Of the 50 respondents (60.2%) who declined a termination, 77.6% gave the reason as ‘termination of pregnancy is forbidden by religion’. The Fatwa Committee, at the Council’s 52nd Muzakarah (Conference) in July 2002, decreed that termination of pregnancy before 120 days is permissible if the foetus is disfigured, ill, and can harm the life of the mother. Yet, as a foetus affected with thalassaemia major does not fully fulfil the above prerequisites, this affects the Malay Muslims’ full acceptance of termination of thalassaemic foetuses and is generally open to interpretation by the priest in question.

Meanwhile, religious leaders of Iran, Saudi Arabia, and the UAE affirmed that married couples should be screened by law for thalassaemia carriers, and the status should be mentioned on the marriage certificate. As a result, more than 100,000 individuals planning to marry were screened for β-thalassaemia. After screening, 90% of couples at-risk (i.e., made up of two carriers) decided not to marry each other. As a result, no new case of thalassaemia was found in the given population. This 90% compliance is in stark contrast with early results for similar carrier detection schemes in Greece, where carrier couples married and produced at-risk pregnancies regardless of an inability at the time to detect and terminate affected pregnancies [92].

### 4.3. Reliable Data for Reliable Disease Control

Nationwide data may be misleading; disparities in the national wealth distribution and regional differences in carrier frequencies call for stratified data to allow for meaningful data-guided policies. Even where nationwide data are informative, the data availability may be limiting. Many factors not captured in disease databases or even well outside the medical sphere influence the progress in disease management and prevention for haemoglobinopathies. Data that would be informative here would be the country-wide average life expectancy of thalassaemia patients or the national budget’s total economic burden due to thalassaemia. Additionally, a less direct but nevertheless significant impact on the success of any screening or prevention programme will be exerted by factors that, for the GGN categories, may have a high level of intragroup variability. These include ethnic and religious diversity, the population size, centrality of the government, type of government (secular, religious, democratic, non-democratic), and overall safety and social stability. Among their other effects, these factors will affect the income of a country and how reliably disease control programmes can be set up and maintained so that only factors not following category definitions may be critical to the success or failure of national disease control. When implementing potentially suitable prevention and disease management strategies, all these factors will facilitate improved downstream plans for the GGN.

The limitations of the research presented in this report include factors that could be mediators in real-time scenarios where governments and officials affirm their priorities in their policymaking concerning treating, preventing, and networking for haemoglobinopathies. This could be inclusive of quantitative data based on homogeneous factors (societal and biological homogeneity) in populations and groups. While the scarcity of data still prevents prediction analysis and forecasting estimates for future cases globally, the classification criteria and country data laid out here will help gather and curate the required data locally and globally among GGN countries for future downstream applications to improve the management of haemoglobinopathies. There is a vital need to establish standard operating procedures (SOPs) to develop registries in all GGN countries and collect and curate data under the GATHER statement (Guidelines for Accurate and Transparent Health Estimates Reporting) [93]. Importantly, local and global registries with a cloud system are needed for future efforts of the GGN.

## 5. Recommendations and Conclusions

Thalassaemia is among the most common monogenetic disorders worldwide and has become a global health burden in affected populations, especially in countries with limited-resource settings. Genomics can help to improve the quality of management and treatment of haemoglobinopathies and thalassaemia patients worldwide, as advancement in genomic medicine can enhance the safety, early timing, and diagnostic accuracy in thalassaemia management. The establishment of an organisation such as the GGN facilitates bridging the gap between developed and developing countries, which may help improve the quality of people’s health, especially in the management and treatment of thalassaemia patients. Networking and collaboration among experts in member countries can assist with policymaking for better health management. In this vein, the criteria proposed and recommended in this study can help the GGN to narrow the gap between these countries.

Moreover, the quality indicators listed in this study can be used for future improvement of disease management among countries within or between groups. Even though the quantitative parameters discussed in this study are related to individual countries, the current data are inadequate to perform a statistically rigorous analysis for future estimates. Therefore, there is a need to develop a common platform to curate and gather data under the proposed criteria in the GGN countries for future undertakings and estimates to tackle the burden of haemoglobinopathies. Population diversity, as an integral part of the prevention and control of haemoglobinopathies, is also advocated. Through this network, each country member can learn and share its experience, resources, and capacities. It is envisaged that the health system in LMICs will be enhanced by genomic knowledge, leading to an understanding of genetic diseases at the molecular level. Prevention programmes, such as prenatal and antenatal screening programmes with a critical dependence on establishing a thalassaemia registry, play essential roles in reducing affected haemoglobinopathy births and improving the treatment and management of patients. Hence, the GGN is a consortium that assists in narrowing the gap between these countries with effective thalassaemia management and prevention programmes by using knowledge and skill transfer under the rubrics of improvement as proposed.

To the best of our knowledge, this is the first report proposing a systematic stratification of countries for their management of thalassaemia, aiming towards data-driven improvements to their prevention, treatment, and database development efforts, for the prevention of thalassaemia across several countries.

## Figures and Tables

**Figure 1 jpm-12-00552-f001:**
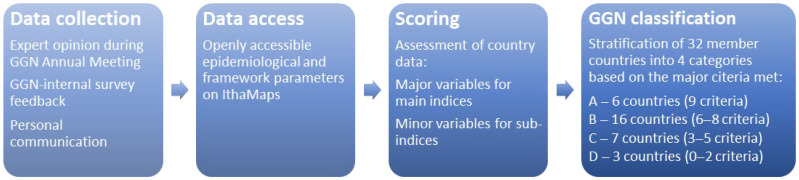
Methodology of the study.

**Figure 2 jpm-12-00552-f002:**
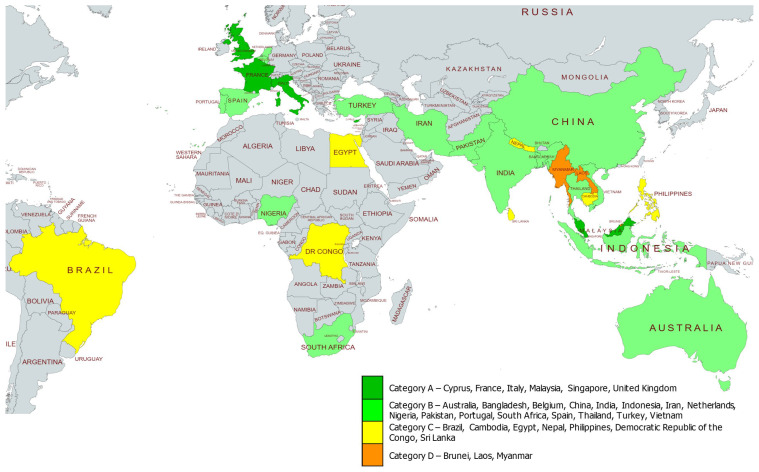
Distribution of the four GGN categories across countries of the GGN consortium.

**Table 1 jpm-12-00552-t001:** Variables in ITHANET database and their categories.

**Qualitative Variables in ITHANET**
*Major Qualitative Variables (Availability of the following parameters)*
1. Genetic counselling
2. Haemoglobinopathies patient registry
3. Dedicated treatment centres
4. Blood transfusion availability
5. Iron chelation availability
6. Prevention programme
7. Prenatal screening
8. Antenatal screening
9. Patient associations
*Minor Qualitative Variables (Availability of the following parameters)*
1. MRI facilities
2. SCD or thalassaemia newborn screening
**Quantitative Variables in ITHANET**
1. Prevalence of β-thalassaemia carriers
2. Prevalence of α-thalassaemia carriers
3. Expected incidence of β-thalassaemia
4. Mutation frequencies
5. Known sickle cell disease patients
6. Prevalence of sickle cell disease carriers
7. Incidence of sickle cell disease
8. Prevalence of Hb E carriers
9. Prevalence of Hb C carriers
10. Known β-thalassaemia patients
11. Incidence of β-thalassaemia

**Table 2 jpm-12-00552-t002:** Categories of GGN countries and country listing with short category descriptions, scoring criteria, and accumulative scoring label.

Category	Short Description	Criterion for Group Scoring	GGN Member Countries	Accumulative Scoring *
A	Countries where services are well-established with a national system for prevention and control	All 9 major qualitative variables are present	Cyprus	A^0^
			France	A^1,2^
			Italy	A^1,2^
			Malaysia	A^1^
			Singapore	A^0^
			United Kingdom (UK)	A^1,2^
B	Countries with efforts to create a partial/fragmented national control programme in place, but with limited availability/accessibility	1–3 major qualitative variables are absent or data are not available	Australia	B^1^
			Bangladesh	B^0^
			Belgium	B^2^
			China	B^0^
			India	B^2^
			Indonesia	B^1^
			Iran	B^0^
			Netherland	B^1,2^
			Nigeria	B^0^
			Pakistan	B^0^
			Portugal	B^1^
			South Africa	B^1^
			Spain	B^2^
			Thailand	B^1^
			Turkey	B^1^
			Vietnam	B^0^
C	Countries where expertise on haemoglobinopathy data collection and management exists but is not part of a sustainable national control programme	4–6 major qualitative variables are absent or data are not available	Brazil	C^1^
			Cambodia	C^0^
			Egypt	C^0^
			Nepal	C^0^
			Philippines	C^2^
			Dem. Rep. of the Congo	C^1,2^
			Sri Lanka	C^0^
D	Countries where expertise and infrastructure for haemoglobinopathy data collection and management are limited	>6 major qualitative variables are absent or data are not available	Brunei	D^0^
			Laos	D^0^
			Myanmar	D^0^

* Minor variables are interpreted with specific reference to routine analyses in the context of SCD and β-thalassaemia. Both the published absence of minor qualitative variables and absence of corresponding information on ITHANET are interpreted as indicating the absence of the minor qualitative variables.

## Data Availability

Not applicable.

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
