# Peer review of "Global Globin Network Consensus Paper: Classification and Stratified Roadmaps for Improved Thalassaemia Care and Prevention in 32 Countries"

_jpm, 2022, doi:10.3390/jpm12040552_

Round 1

Reviewer 1 Report

A very interesting study that proposes a universally applicable system to evaluate and group countries based on qualitative indicators according to the quality of care, treatment, and prevention of hemoglobinopathies.

For the introduction, I would propose to enrich the bibliography, looks very poor with 9 citations. 

Regarding the Material and Method section, I would like to be added a figure presenting the procedure in order to be more simple for the readers to understand it. 

Also, in the results section the tables could be imrpoved....

It is well written and the conlcusion very clear!

Author Response

Response to reviewer 1 comments:

Point 1: A very interesting study that proposes a universally applicable system to evaluate and group countries based on qualitative indicators according to the quality of care, treatment, and prevention of hemoglobinopathies.

Response 1: We thank the Reviewer for the favourable evaluation.

Point 2: For the introduction, I would propose to enrich the bibliography, looks very poor with 9 citations.

Response 2: The revised Introduction is now supported by a total of 20 references, and we hope that this addresses the point satisfactorily.

Point 3: Regarding the Material and Method section, I would like to be added a figure presenting the procedure in order to be more simple for the readers to understand it.

Response 3: Thank you for suggesting this and making the manuscript more accessible to the reader. We have introduced a horizontal flow chart for the classification process and think that it clearly illustrates the procedure.

Point 4: Also, in the results section the tables could be imrpoved....

Response 4: We have streamlined the tables by introducing uniform abbreviations across tables and removing some of the less pertinent information. We hope that this makes the tables more comprehensible.

Point 5: It is well written and the conlcusion very clear!

Response 5: Thank you so much for this positive assessment and for the helpful suggestions.

Note: Please see the attachment for the corrected manuscript. 

Reviewer 2 Report

This article ‘Global Globin Network Consensus Paper: Classification and 2 Stratified Roadmaps for Improved Thalassaemia Care and Prevention in 31 Countries’ reports concise and precise parameters and scoring criteria for categorizing 31 countries into 4. Brief of all the countries disease management will be a resource to get success in recommendations, improvement and prevention. I have only two minor comments.

In materials and methods, please put ‘a’ and ‘ccumlative’ together.

In Figure 1 the legend category A doesn’t match to its color in the map. Please correct it.

Author Response

Response to reviewer 2 comments:

Point 1: This article ‘Global Globin Network Consensus Paper: Classification and 2 Stratified Roadmaps for Improved Thalassaemia Care and Prevention in 31 Countries’ reports concise and precise parameters and scoring criteria for categorizing 31 countries into 4. Brief of all the countries disease management will be a resource to get success in recommendations, improvement and prevention. I have only two minor comments.

Response 1: We thank the Reviewer for this positive assessment and hope indeed that classifications and recommendations in this article will facilitate the work of others and will help improve care and prevention of thalassaemia.

Point 2: In materials and methods, please put ‘a’ and ‘ccumlative’ together.

Response 2: Done. [current lines 178 & 180 “a cumulative” -> “an accumulative”]

Point 3: In Figure 1 the legend category A doesn’t match to its color in the map. Please correct it.

Response 3: Thank you for the vigilance. We have revised the colour scheme of the, now Figure 2, for a more intuitive perception of classifications. In the process we have taken care to have identical colouring in map and legend. We hope that this addresses this point adequately.

Note: Please see the attachment for the corrected manuscript. 
